# Proton Dynamics of Water Diffusion in Shrimp Hydrolysates Flour and Effects of Moisture Absorption on Its Properties

**DOI:** 10.3390/foods10051137

**Published:** 2021-05-20

**Authors:** Yue Zhao, Songyi Lin, Ruiwen Yang, Dong Chen, Na Sun

**Affiliations:** 1National Engineering Research Center of Seafood, School of Food Science and Technology, Dalian Polytechnic University, Dalian 116034, China; zhaoyue1268@163.com (Y.Z.); linsongyi730@163.com (S.L.); chendong689689@126.com (D.C.); 2College of Food Science and Engineering, Jilin University, Changchun 130062, China; maomaoyrw@163.com

**Keywords:** moisture absorption, volatile compounds, radicals scavenging, low-field nuclear magnetic resonance, dynamic vapor sorption

## Abstract

Moisture absorbed into shrimp hydrolysates (SHs) flour profoundly affected its properties. The unstored hydrolysate flour was called SHs-0h and SHs stored for 30 h at 25 °C and 75% relative humidity was named SHs-30. During the process of storage, the moisture dynamics in SHs flour were investigated by dynamic vapor sorption (DVS) and low-field nuclear magnetic resonance (LF-NMR). The effects of moisture absorption on the radicals scavenging rates of SHs flour were evaluated by electron paramagnetic resonance (EPR). The effects of moisture absorption on secondary structure were studied by mid-infrared (MIR) spectroscopy and infrared microimaging spectroscopy. The changes of volatile components were monitored by purge and trap coupled with gas chromatography-mass spectrometry (PT-GC-MS). DVS results showed that the moisture absorption rate of SHs flour could reach a maximum of 88.93%. Meanwhile, the water was transformed into more stable water with shorter relaxation times. The porous structure of the SHs-30 h flour disappeared and became smoother compared to SH-0 h flour. DPPH (31.09 ± 0.54%) and OH (26.62 ± 1.14%) radicals scavenging rates of SHs-30 h significantly reduced (*p* < 0.05) compared to that of SHs-0 h flour. The vibrations of the MIR absorbance peaks were changed. Finally, eight volatile components disappeared and six new volatile compounds were found. This study provided a theory basis for moisture dynamics in peptide flour during the storage process.

## 1. Introduction

Food proteins, especially hydrolyzed peptides, have important physiological activities including immunomodulatory, anticancer, and antioxidant [1,2,3]. Bioactive peptides are considered natural ingredients, and the preparation process is relatively simple and has no side effects [4]. In recent years, bioactive peptides have become more and more popular due to their functional properties. A study reported that collagen peptides have antioxidant and antitumor activities [4]. However, peptide flour was vulnerable to the moisture, which induced the changes of the peptide characters, such as sticky, agglomeration, and liquefaction. The hygroscopic theory is that water migrates from the high water activity region to the low water activity region until the system of materials reaches equilibrium of water [5]. After casein was stored for a year, the odor of the product changed noticeably with accompanying by an unpleasant odor [6]. Meanwhile, the nutrients losses and the deterioration of food quality are often developed in the peptide flour [7]. The appropriate processing and storage conditions should be explored to keep the good quality of peptide flour.

Recently, the characteristics of water distribution in peptide flour were elucidated during the process of storage [7,8]. There were four categories of water in peptide flour and moisture absorption capacity of peptides was related to molecular weight. Yang et al. monitored the changes of the egg white peptide flour during the moisture absorption, and verified that the distribution of proton and certain properties were changed by moisture absorption [9]. Four categories of different water existed in different phases. Lin et al. revealed that moisture absorption affected the morphology and structure of peptides and deliquescent peptide flour generated new volatile compounds [10]. Obviously, moisture absorption can affect the physical and chemical properties of peptide flour. However, the study on the hygroscopicity of peptides is not in-depth enough.

The dynamic vapor sorption (DVS) can be used to measure the equilibrium moisture contents of materials, such as shiitake mushroom and soybean peptide flour, in varying relative humidity (RH) [7,11]. In the present study, water sorption kinetics of shrimp hydrolysates flour in 0–95% RH was monitored by DVS to estimate the water adsorption capacity. Low-field nuclear magnetic resonance (LF-NMR) was used to monitor the water distribution and understand interactions between water and peptides or proteins by monitoring the proton dynamics [12,13,14]. The physicochemical properties of unstored hydrolysate flour and stored hydrolysate flour for 30 h at 25 °C and 75% RH were investigated by electron paramagnetic resonance (EPR) and mid-infrared (MIR) and infrared microimaging spectroscopy technology. Meanwhile, the changes of volatile compounds in shrimp hydrolysates flour were also identified and quantified by trap coupled with gas chromatography-mass spectrometry (PT-GC-MS). The objectives of this study were to trace water dynamics of shrimp hydrolysates flour and the effects of moisture absorption on its properties. These results provided a valuable theoretical basis for the storage and application of the peptides flour.

## 2. Materials and Methods

### 2.1. Reagents and Chemicals

Two kilograms of shrimp meat of freshwater shrimp were purchased at a local fish market in Dalian (China). The 2.4 L of Alcalase was obtained from Novozyme (Bagsvaerd, Denmark). The 5,5-dimethy-1-pyrroline-N-oxide (DMPO) and potassium bromide (KBr) powder were obtained from Sigma Chemicals Co. (Madison, WI, USA). All other chemicals were analytical grade and available from Peking Chemical Plant (Beijing, China).

### 2.2. Preparation of Shrimp Hydrolysate (SHs) Flour by Alcalase

The shrimp hydrolysate flour was obtained with the method of Zhao et al. with appropriate modifications [15]. First, the shrimp meat was smashed by pulverizer for 5 min. 100 g of smashed shrimp meat was added to 400 mL of deionized water and stirred with stirring paddle. The solution was heated to 90 °C and kept at 90 °C for 10 min. In addition, then the temperature of solution was reduced to 50 °C. 2 mL Alcalase was added to the solution. Then, the solution was kept at 50 °C and pH 8.5 for 3 h. During the process of enzymatic hydrolysis, the solution was constantly stirred. After that, the hydrolysate solution was kept at 90 °C for 10 min and centrifuged at 14,400× *g* for 10 min at 4 °C and the pH of supernatant was adjusted to 7.0. Finally, the hydrolysate was stored at −80 °C and freeze-dried for further use.

### 2.3. Dynamic Vapor Sorption (DVS) Measurements in RH from 0% to 95%

Moisture sorption experiment was performed according to the method of Young et al. at 25 °C by exposing the SHs flour to different values of RH within the range from 0 to 95% using the DVS (Surface Measurement Systems Ltd., London, UK) [16]. To understand the hygroscopicity of hydrolysate flour under different relative humidity (RH) conditions, DVS instrument was used to monitor the hygroscopicity. The running program was set on the instrument’s computer. RH ranged from 0% to 95%, and then from 95% to 0%. The experiment was conducted in two cycles. The RH was raised in RH steps of 10% from 0% to 90% RH and in RH steps of 5% from 90% to 95%. It increased to the next target value if the sample mass rate with time was lower than 0.002% min-1 or the running time was over 360 min. The container was weighed and mass of container was removed. Then, 13 mg of SHs flour was placed in a container of the instrument and the program was run. First, the SHs flour was dried by exposing it to dry nitrogen until a constant weight was reached. The total mass of the sample and container was recorded. RH started to change until the program was finished.

### 2.4. Low-Field Nuclear Magnetic Resonance (LF-NMR) Test

Next, high RH conditions (75% RH) was selected to conduct LF-NMR test, in order to better understand the proton dynamics of water diffusion. The LF-NMR experiment was performed according to the method of Yang et al. with some modifications [8]. The water dynamics and distribution of SHs flour were monitored by LF-NMR equipment. The NMI20-030H-INMR analyzer (Niumag Electric Corp., Shanghai, China) was used in the LF-NMR experiment. Then, 1.840 g of SHs flour was placed in the sealed glass bottle and weighed, then placed into the nuclear magnetic tube for testing. The sample was kept in clean and relatively closed LHS-250SC box (Shanghai bluepard instruments Co., Ltd., Shanghai, China) with constant temperature and humidity. The temperature and relative humidity (RH) were set at 25 °C and 75%, respectively. During the process of storage (from 0 to 30 h), the unsealed glass bottle was left in the box in the experimental environment. Next, the glass bottle was sealed and the sample was weighted and taken photos, when the sample had adsorbed water for 0, 2, 4, 6, 8, 10, 20, and 30 h. Transverse relaxation, *T*_2_, was monitored by the Carr-Purcell-Meiboom-Gill (CPMG) sequence under the conditions of 21 MHz proton resonance frequency, 200 kHz spectral width (SW), 200 ms half-echo time t-value (time between 90° pulse and 180° pulse), the number of sampling (TD) of 400026, the number of repetitive scans (NS) of 12, and the number of echoes of 10,000. When the sample had been stored for 0, 2, 4, 6, 8, 10, 20, and 30 h, it was inside an NMR glass tube. Then, the sample was performed nuclear magnetic imaging (MRI) experiments using spin echo sequence with the sampling number (TD) of 12, the spectral width (SW) of 100 kHz, and the repetitive scans number (NS) of 8. The data were processed with Multi Inv Analysis soft (Niumag Electric Corp., Shanghai, China). The *T*_2_ curve was substituted into the relaxation model by the iterative optimization method Equation.
(1)$$M\left(t\right)=\overset{n}{\underset{i=1}{\sum }}A_{2i}\;esp\left(\frac{-t}{T_{2i}}\right)+d\left(t\right)$$
where *M*(*t*) is the residual magnetization at a given time t after application of the first RF pulse, n is the quantity of the exponential functions in the specimen, *A*_2*i*_ and *T*_2*i*_ is the signal intensity of relaxation amplitude and time of the ith component, and *d(t)* is the residual error. The unstored hydrolysate flour was named SHs-0 h and stored SHs for 30 h at 25 °C and 75% RH was named SHs-30. SHs-0 h and SHs-30 h were selected for subsequent.

### 2.5. Detection of Morphological of SHs-0 h and SHs-30 h Flour by Scanning Electron Microscopy (SEM)

The SHs flour (1.840 g) was placed in the glass bottle under 75% RH conditions to facilitate water absorption for 30 h at 25 °C. The glass bottle was kept in clean and relatively closed LHS-250SC box with constant temperature and humidity. The unstored SHs flour was named SHs-0 h and stored SHs flour for 30 h was named SHs-30 h. The effects of moisture absorption on morphological changes of SHs flour were evaluated by SEM. SEM experiment was performed according to the previous method described by Zhao et al. [15]. The SHs-0 h and SHs-30 h flour were observed with a JSM-6700F field emission SEM (JEOL Ltd., Tokyo, Japan) by the E-1045 ion beam sputtering instrument (Hitachi Co. Ltd., Japan). The SHs-0 h or SHs-30 h flour (1 mg) was adhered to a circular aluminum specimen and photographed under the condition with a potential of 3000 V. The SEM micrographs magnification was 1000 and 5000×.

### 2.6. Detection of Radicals of SHs-0 h and SHs-30 Flour by Electron Paramagnetic Resonance (EPR)

The SHs flour (1.840 g) was placed in the glass bottle under 75% RH conditions to facilitate water absorption for 30 h at 25 °C. The glass bottle was kept in clean and relatively closed LHS-250SC box with constant temperature and humidity. The unstored SHs flour was named SHs-0 h and stored SHs flour for 30 h was named SHs-30 h. The effects of moisture absorption on radicals of SHs flour were evaluated. The radicals were measured by EPR spectrometer (Bruker, Karisruhe, Germany) and the 2,2-diphenyl-1-picrylhydrazyl (DPPH) and hydroxyl (OH) scavenging capacity were measured based on the method of Ding et al. [17]. The concentration of SHs flour was 1mg/mL. The OH radical scavenging activity was measured in the system with 38 μL of PBS (0.15 mol/L), 39 μL of solution of SHs flour, and 5 μL of DMPO. In addition, then 10 μL of EDTA-2Na-Fe^2+^ (6 mM) and 8 μL of H_2_O_2_ (6%) were added the system of mixture. The mixed solution was heated in 40 °C for 30 min and then was sucked into polytetrafluoroethylene capillary tube for the detection of OH radical scavenging activity. The equipment parameters are as follows: Sampleg-factor (2), Center filed (3441.95 G), Sweep width (200 G), Receiver gain (40 dB), Number of scans (3) Attenuation (25 dB). The DPPH radical scavenging activity was measured in the system with 40 μL (0.1 mol/L) of PBS, 40 μL of DPPH (200 μmol/L DPPH in 95% ethanol), and 20 μ of solution of SHs flour. The solution was placed in the dark condition for 30 min and was centrifuged for 10 min (2000× *g*). The DPPH radical scavenging activity was measured through the EPR. The equipment parameters are as follows: Sampleg-factor (2), Center filed (3440 G), Sweep width (100 G), Receiver gain (40 dB), Number of scans (3), Time constant (40.96 ms), Attenuation (10 dB). The radical scavenging activity was obtained by following equation:(2)$$\mathrm{Radical}\;\mathrm{scavenging}\;\mathrm{activity}\;\left(\%\right)=\frac{\mathrm{Ac}-\mathrm{As}}{\mathrm{Ac}}\times 100\%$$

The Ac was the curve integral area of the control group (distilled water group) and as was the curve integral area of the SHs flour group.

### 2.7. Mid-Infrared (MIR) Spectroscopy Analysis of SHs-0 h and SHs-30 Flour

The SHs flour (1.840 g) was placed in the glass bottle under 75% RH conditions to facilitate water absorption for 30 h at 25 °C. The glass bottle was kept in clean and relatively closed LHS-250SC box with constant temperature and humidity. The unstored SHs flour was named SHs-0 h and stored SHs flour for 30 h was named SHs-30 h. DPPH and OH scavenging capacity were associated with changes in functional groups. Thus, the effects of moisture absorption on secondary structure were evaluated by MIR spectroscopy analysis. The MIR analysis was performed according to the method of Lin et al. [18]. MIR spectroscopy was used to study the structure changes of SHs-0 h and SHs-30 h flour. KBr was dried at 130 °C for 5 h. The 2 mg of dried SHs-0 h or SHs-30 h flour was mixed with 200 mg of potassium bromide and the potassium bromide and SHs-0 h and SHs-30 h flour pellets were prepared for measure. MIR spectra were measured using an IR Prestige-21 Fourier transform infrared spectrometer (Perkin Elmer, Shanghai, China) at a resolution of 4 cm^−1^. The all MIR spectra were baseline corrected and MIR spectra of SHs- 0 h and SHs-30 h flour were obtained.

### 2.8. Infrared Microscopic Imaging of SHs-0 h and SHs-30 Flour

The SHs flour (1.840 g) was placed in the glass bottle under 75% RH conditions to facilitate water absorption for 30 h at 25 °C. The glass bottle was kept in clean and relatively closed LHS-250SC box with constant temperature and humidity. The unstored SHs flour was named SHs-0 h and stored SHs flour for 30 h was named SHs-30 h. Infrared microscopic imaging technology was used to further study the effects of moisture absorption on secondary structure and can be used to measure the functional group information and spatial distribution information of the molecular structure of proteins and other samples. Infrared microscopic imaging was performed according to the method of Bonwell et al. with some modifications [19]. The SHs-0 h or SHs-30 h flour (2 mg) was placed on the platform and then three points were selected in the sample area for repeated determination to obtain the microscopic infrared imaging. Detection wavelength: 4000–750 cm^−1^; Interference speed: 1.0 cm/s; Resolution: 4 cm^−1^; Pixel. Finally, the wavelength range was selected and the microscopic infrared imaging was obtained. 

### 2.9. Identification and Quantification of the Volatile Compounds of SHs-0 h and SHs-30 h Flour by PT-GC-MS

The SHs flour (1.840 g) was placed in the glass bottle under 75% RH conditions to facilitate water absorption for 30 h at 25 °C. The glass bottle was kept in clean and relatively closed LHS-250SC box with constant temperature and humidity. The unstored SHs flour was named SHs-0 h and stored SHs flour for 30 h was named SHs-30 h. Volatiles components in SHs-0 h and SHs-30 h flour were collected by purge and trap coupled with gas chromatography-mass spectrometry (PT-GC-MS) (Agilent/Atomx/7890B/5977A) according to the method of PozoBayón et al. [20]. HP-5MS Ultra Inert (30 m × 0.25 mm × 0.25 μm; Agilent 19091S-433UI) was selected as the chromatographic column. The 0.5 g of SHs-0 h or SHs-30 h flour was used for research. Parameters of purge and trap are as follows: sample vial temperature (50 °C), purge time (40 min), purge flow (40 mL/min), purge temperature (20 °C). The injection mode was split with a 10:1 ratio. The 35 °C was selected as the initial temperature of the column oven and the condition was kept for 3 min. Subsequently, the temperature increased to 280 °C at a rate of 5 °C /min for 10 min. The running time was maintained for 62 min. Helium was selected as the carrier gas and the pressure was kept constant at 46.73 kPa. The column flow rate was kept at 14 mL/min. The mass scan range of m/z was kept at 35 and 400 amu. The MS ion source temperature and quard temperature were controlled 230 °C and 150 °C, respectively. Quantification of volatile compounds was obtained by the retention index (RI). The RI value was compared with value in the Linear Retention Indices (LRI) database. When the R. match value is higher, the compound match is more accurate. The analysis was performed according to previous values and the corresponding mass spectrum in a database. The standard method was used for calculating peak areas and concentrations of volatile compounds. Finally, the hierarchical cluster analysis and principal component analysis (PCA) were conducted.

### 2.10. Statistical Analysis

Statistical analyses were performed using the SPSS 21.0 version software (SPSS Inc., Chicago, IL, USA). The experimental results were shown as means ± standard deviation (SD). One-way analysis of variance with Least-significant difference (LSD) test was used to analyze significant difference. Three replicated measured results were performed and a level of significance *p* < 0.05 was considered significant.

## 3. Results and Discussion

### 3.1. Water Vapor Sorption Analyses of SHs Flour in RH from 0% to 95%

Peptide flour as a hygroscopic material always contains a certain percentage of water and continuously exchanges with the environmental moisture by absorbing moisture [21]. Peptide flour absorbs moisture when atmospheric RH is high (adsorption) and releases moisture when atmospheric RH is low (desorption). With the step-by-step increase or decrease of RH, the moisture content in the samples changed, approaching a constant value for a long time. The value is defined as the equilibrium moisture content. The DVS was used to evaluate the hygroscopicity of SHs flour. The water sorption kinetic curve was shown in (Figure 1A). The change of water content was presented by the change in the weight of the SHs flour. When a new target RH value was set, the water content of the SHs flour gradually changed during the process of adsorption and desorption until the state of the sorption reached a new state. After the experiment condition reached, the RH changed to the next new value. The SHs flour, after preparation, contained a small amount of water and it was dehydrated to equilibrium at 0% RH. The maximum equilibrium time was set to 360 min. During the process of adsorption, with the adsorption time increasing, the water content of SHs flour gradually increased. When the RH reduced, the moisture of SHs flour was desorbed. After the RH reached 95%, the process of desorption started to run. Although the RH had dropped 90%, the weight of SHs flour continued to increase. In this stage, the absorbed moisture rate of the SHs flour was higher than the moisture loss rate. The water content reached 83.36% at 95% RH and the maximum value was 88.93% at 90% RH in the phase of desorption. These indicated that the change in SHs flour weight did not reach the equilibrium at 95% RH. The weight of SHs flour decreased rapidly when the RH was less than 90% until the RH dropped to 20%. Similarly, the SHs flour adsorption in cycle 2 also needed plenty of time. The weight of SHs flour decreased 0.73% from 0% to 20% RH during the process of adsorption in cycle 2. These revealed that the SHs flour absorbed moisture rate was lower than the moisture loss rate in this stage. In particular, the time that water content reached the maximum water content was particularly short at 20% and 30% RH and the equilibrium times were 58 min and 88 min, respectively. Similarly, when the moisture was desorbed at 90% RH, the moisture content reached the maximum value and the weight of SHs flour slowly increased in limited time. As the kinetic curve of SHs flour is shown in (Figure 1A), the SHs flour slightly absorbed moisture during the beginning of experiment (10% and 20%). When the RH exceeded 20%, the increasing rate of weight quickly increased until the RH reached 95%.

According to the water sorption kinetic curve, the sorption and desorption isotherms plots were represented in (Figure 1C). The water sorption kinetic curve showed changes in weight with increasing slowly in the beginning of the experiment. When the RH gradually increased from 10% to 50%, the weight changes changed from 0.22% to 8.51%. Subsequently, the weight of SHs flour changed rapidly. The SHs flour contained a lot of peptides and the potential hydrogen bonding sites might exist. In a high RH, the polar groups of the peptides bound to the water and the water around the peptides may continue to absorb other water molecules. Moisture absorption occurred within the SHs flour and the particle amorphous state of SHs flour became a partially crystalline hydrate (Figure 1B). For the experiment of amorphous materials, in most situations, the sorption isotherm curve was below the isotherm curve of desorption, but for crystalline materials, the opposite is true [22]. In the adsorption process of cycle 1, the absorption moisture destroyed the loose structure of SHs flour. The SHs flour became hydrated with a large amount of water. The sorption and desorption isotherms in cycle 1 were not close. This might be due to the combination of water and peptides in SHs flour and the water combined with SHs flour could not be desorbed. There were a lot of differences in sorption isotherms cycle 1 and cycle 2. Water combined with peptides might affect water absorption. The isotherms of desorption in cycle 1 and cycle 2 were almost coincident. The DVS was used to evaluate the hygroscopicity of SHs flour in different RH and discussed the changes during the process of moisture adsorption and desorption. Then, the SHs flour was stored in a high and high RH and the water migration in SHs flour was discussed.

### 3.2. Water Distribution of SHs Flour Monitored by LF-NMR during Process of Storage

The SHs flour was stored for 30 h in a high RH (75%) and the water migration in SHs flour was studied. As shown in (Figure 2A), the SHs flour was photographed during the process of moisture adsorption under the conditions of 75% RH and 25 °C. When the SHs flour did not performed moisture absorption at the standing time 0 h, the surface of SHs flour was the amorphous and loose state. When the storage time reached 10 h, the loose state was slightly damaged and the color was slightly deeper. The deliquescence induced the changes on the surface of SHs flour at 10 h. When the storage time was 20 h, the surface shape of SHs flour generated obvious changes. Compared with SHs-0 h, the color of SHs flour stored for 20 h (SHs-20 h) was greatly deeper, the volume was obviously smaller, and the surface became sticky. When the storage time increased to 30 h, the color of SHs flour stored for 30 h (SHs-30 h) was the deepest; the volume of sample was the smallest because of the action of water tension [10], and the surface of sample was obviously liquefied. The amorphous and loose state of SHs flour changed into a partially crystalline state, and the volume of SHs flour dramatically shrunk. If time is long enough, the peptide flour may be completely liquefied. Moreover, the mass of sample significantly increased throughout the experiment (*p* < 0.05), which indicated that the SHs flour absorbed plenty of water. Then, the nuclear magnetic imaging (MRI) technology was used to explain the signal of hydrogen protons in order to further understand the changes of water in SHs flour [23].

The MRI technology can monitor the changes of the free water in food by identifying the signal of hydrogen protons. Under the condition of 75% RH and 25 °C, the hydrogen proton density of SHs flour was detected by the MRI system. The pseudo-color images were shown in (Figure 2B) which red and blue colors mean high and low proton density, respectively [24]. The signal of hydrogen proton cannot be observed in the early stages of the experiment. With the increase of moisture adsorption time, the color of the pseudo-color images gradually deepened and the signal of the hydrogen proton density in the SHs flour strengthened. During the process of storage, the map of proton density showed that the moisture gradually spread into a circular structure and then diffused inward from the outside. When moisture adsorption time reached 30 h, the map of proton density showed obvious differences. These indicated that more and more water diffused into the inward of the SHs flour to promote the enhancement of the signal and color. However, the detailed information about changes of the proton signal cannot be observed from the MRI [25]. The changes of the proton signal of SHs flour were further studied by relaxation time distribution curves of LF-NMR.

In general, the water dynamics in food were obtained by LF-NMR technology [25,26]. The interaction of water and peptides was widespread, and the interactive process was inferred to be a complex system. The water dynamics and distribution of SHs flour during the process of moisture absorption were investigated by LF-NMR. In most LF-NMR analyses, *T*_2_ values can be divided into three components including *T*_21_, *T*_22_ and *T*_23_. First, *T*_21_, component less than 10 ms, includes the protein-associated water, secondly, *T*_22_, the second component ranging from 10–100 ms, referring to the immobilized water and thirdly, *T*_23_, a component ranging from 100–200 ms referring to the free water [13,27,28,29]. The relaxation time distribution curves of SHs flour at 75% RH and 25 °C were shown in (Figure 2C). Three hydrogen proton fractions in the process of moisture absorption were observed, revealing that three categories of water were observed in the SHs flour [30]. We found that the SHs flour had varying relaxation times and amplitudes during the process of storage. Studies reported that *T*_2_ relaxation consists of three components, each of which may be determined as follows: *T*_21_, the first component between 1 and 10 ms representing bound water, *T*_22_, the second component between 10 and 100 ms reflecting immobilized water. *T*_23_, defined as free water, was the slowest component with a relaxation time ranging from 100 to 500 ms. This was consistent with our results. Three components of water bind tightly to large molecules [31]. With the increasing storage time, *T*_22_ and *T*_23_ first were observed an increasing trend, and when stored for 30 h, they decreased to the minimum, which indicated that the binding force between peptides and bound water was strengthened [32]. Study has shown that the binding force between material and bound water was strengthened when *T*_22_ became smaller [33]. As shown in (Figure 2C,D), the content of bound and immobilized water gradually increased throughout the experiment. In the initial stage of moisture absorption, the immobilized water migrated to the bound water. After the moisture absorption time was 20 h, the bound water significantly decreased (*p* < 0.05) and migrated to the immobilized water, which also indicated that the combination of water and peptides in this system was becoming more and more unstable. With the time increasing, the corresponding signal amplitude of bound and immobilized water significantly decreased. The SHs flour was vulnerable to moisture absorption and signal amplitude between bound and immobilized water migrated or exchanged. There were many intermolecular effects such as hydrogen bonds, electrostatic interactions, or electron density repulsions in protein structure [34]. On study showed that the behaviors of moisture absorption were related to the polar groups of the peptides [9]. The interactions of water and peptides were related to these intermolecular effects. When proteins were enzymolyzed, short peptides were produced [35]. The short peptides might belong to the possible hydrogen bonding sites which induced the interactions between the peptides and moisture [10]. These sites probably provided water molecules with attachment sites and increased the chances of the interaction between water and peptides during the process of moisture absorption. During the enzymatic hydrolysis of shrimp meat, amino and carboxyl groups generated and these might induce absorption moisture of SHs flour. The water around the peptides formed the hydrogen bond with the groups of SHs flour. At the initial stage, the groups on the surface of peptides combined with water. At the end of the experiment, SHs flour formed the liquefied surface. It could be seen from LF-NMR results (Figure 2E), the distribution of transverse relaxation times of SHs flour was changed during moisture absorption. The bound and immobilized water accounted for the majority of total water and transformed into each other. Because the migrations of water were complex, the detailed reasons for these water migrations have been still unclear. In-depth research needs to be performed to elucidate the mechanisms of these water migrations. Meanwhile, it revealed that LF-NMR was an effective tool to elucidate water distribution during the process of moisture absorption.

### 3.3. Effects of Moisture Absorption on Morphological Changes of SHs Flour 

The surface morphology and microstructure of SHs-0 h and SHs-30 h flour can be obtained through scanning electron microscopy (SEM). As shown in (Figure 3A), the SEM images of SHs flour were exhibited at magnification factors of 1000 and 5000 fold. The SHs flour before moisture absorption exhibited a porous structure. After the moisture absorption, the surface of SHs flour became smoother and lost its porous structure. In particular, moisture can have a great effect on amorphous materials. Compared with their crystalline solids, amorphous solids can absorb more water, which moisture absorption might result in particle agglomeration and flour caking [36]. The hygroscopicity caused morphological changes of peptide, which was changed from an amorphous structure into different agglomerates [37]. This was consistent with our findings that moisture absorption led to flour agglomeration and caking and lost the porous structure.

### 3.4. Effects of Moisture Absorption on Radicals Scavenging Activity of SHs Flour

Water migration characteristics during the moisture absorption process of SHs flour were investigated. The moisture absorption might affect the physical chemistry properties such as radicals scavenging activity. Theoretically, the peptides can scavenge radicals. Next, the effect of moisture absorption on the antioxidant capacity of the SHs flour was evaluated by EPR measurement. The 1,1-diphenyl-2-picrylhydrazyl (DPPH) and hydroxyl (OH) radical scavenging activity of SHs flour was measured and the EPR signal intensity was shown (Figure 3). The DPPH and OH radical scavenging activity of SHs-0 h and SHs-30 h flour were obtained through the integral of the peak. From the EPR results, the radicals scavenging activity of the SHs-30 h decreased significantly compared with that of the SHs-0 h (*p* < 0.05). A similar study reported that the antioxidant ability of pine nut peptide decreased after the storage (*p* < 0.05) [38]. The inherent moisture of peptides flour increased the oxygen solubility [10]. Thus, the reduction of antioxidant ability after the storage might be due to the increase of oxygen solubility. The reductive groups in hydrolysate flour were oxidized by oxygen and the antioxidant ability of SHs flour showed a certain degree of decline. These indicated that properties of SHs flour were not stable in a high humidity environment. However, the mechanism needs to be further studied.

### 3.5. Changes of Functional Groups after Moisture Absorption

MIR spectra can reflect the vibration of the chemical bonds [39]. The adsorption of proteins was related to functional groups properties. As shown in (Figure 4A), the MIR experiment was used to further study the effects of moisture absorption on secondary structure. In the range of 4000–2500 cm^−1^, vibrations of the absorbance peaks resulting from the hydrogen groups, such as the -OH, -NH_2_ and -NH stretch vibrations (3400–3200 cm^−1^), -CH3 stretch vibrations (2960 ± 5 cm^−1^ and 2870 ± 10 cm^−1^), and -CH2 stretch vibrations (2930 ± 5 cm^−1^). The absorption band at 3400–3200 cm^−1^, amounting to the O-H and N-H bonds, was attributed to water of hydration [40]. A similar study reported that changes in the absorption peaks at 3500–3300 cm^−1^ in MIR spectra reflected changes in the intermolecular hydrogen bonding force of peptides [41]. Hydrogen bonding plays an important role in maintaining the stability of the secondary structure of proteins and peptides. The types of functional groups of the peptides did not change, but the intensity of absorbance peaks was affected by moisture absorption. The change of absorbance peaks at 3500–3300 cm^−1^ of the peptides may indicate that the secondary structure of the peptides was affected by moisture absorption. During the process of moisture absorption, some water which was combined strongly with peptides might not be removed by the method of freeze-dried. This indicated the water might affect -OH stretch vibration. The water bonded to the peptides would increase the adsorption potential for external water in air [10]. From the MIR spectra, the intensity of absorption peaks of peptides was affected by moisture absorption.

Infrared microscopic imaging technology was used to further study the effects of moisture absorption on secondary structure and can be used to measure the functional group information and spatial distribution information of the molecular structure of proteins and other samples. For example, the influence of the distribution of protein flour can be analyzed using the infrared microscopic imaging technology [42]. Infrared microscopic imaging technology was used to detect protein hydrations in living cells [43]. In the MIR spectra, the characteristic absorbance peaks of the secondary structure of the peptides appeared in the amide I band (1700–1600 cm^−1^) amide II band (1560–1535 cm^−1^) [44,45]. Therefore, as shown in (Figure 4B–E), infrared microscopic spectrums of these areas were selected to analyze the influence of moisture absorption on the spatial distribution of peptides. The color intensity of the infrared microscopic spectrum indicates the homogeneity of the sample distribution. Shown in red was the high intensity of the protein [45]. The microscopic spectrogram of SHs-0 h flour showed a pink dot or block distribution, while the spectrogram of SHs-30 h flour showed a pink band-like distribution, which became more concentrated. This result directly reflected the effect of moisture absorption on the spatial distribution of the peptides. In the range of 1700–1600 cm^−1^, vibrations of the absorbance peaks resulting from high intensity of the protein amide I absorption band. In the range of 1560–1535 cm^−1^, vibrations of the absorbance peaks resulting from high intensity of the protein amide II absorption band [46]. These results directly reflected that moisture absorption affected the absorption intensity and distribution of peptides, and did not change the functional groups of peptides. Wang et al. reported that the OH radical scavenging capacity of sea cucumber peptide flour decreased after it was stored for 24 h [27], which was consistent with our result. The secondary structure of peptides was closely related to antioxidant activity, and the change of vibrations of the absorbance peaks may be one of the reasons that the antioxidant activity on peptides decreased [47].

### 3.6. Effects of Moisture Adsorption on Volatile Compounds of SHs Flour

Volatile compounds in SHs-0 h and SHs-30 h flour were identified according to the matched retention index (RI). Most volatile compounds were found in databases, which could represent the authority of the identification. Cyclohexanone served as an internal standard and the peak areas were standardized. The characteristic chromatograms of the SHs-0 h and SHs-30 h are shown in (Figure 5A,B), respectively. As shows in (Table 1), a total of 18 volatile compounds with Chemical Abstracts Service (CAS) registry numbers were identified in the SHs-0 h and SHs-30 h. The volatile compounds were classified according to their structure, including 7 alcohols, 4 aldehydes, 4 arenes, 2 acids, and 1 ester. The SHs-0 h and SHs-30 h contained 12 and 10 major volatile components, respectively. There were four kinds of volatile compounds in SHs-0 h and SHs-30 h including (b) 4-penten-2-ol, (l) benzaldehyde, (n) nonanal, and (r) oleic acid. After SHs flour was stored for 30 h, eight volatile compounds disappeared, which included (a) 2-methoxy-ethanol, (f) 2-pentanol, (g) paraldehyde, (h) butanoic acid, (i) hexanal, (k) o-xylene, (m) 2-ethyl-1-hexanol, and (o) naphthalene. The SHs-30 h was detected 6 new volatile compounds including (c) 2-nitro-ethanol, (d) 1-methoxy-2-propanol, (e) 3-methyl-2-butanol, 2, (b) 4-dimethyl-heptane, (p) tetradecane and (q) 2-hexyl-1-decanol. Cluster analysis of SHs flour’s volatile compounds was conducted according to the standardized peak areas of volatiles. The classification of each flavor component in the SHs-0 h and SHs-30 h flour can be clearly observed. As shown in (Figure 5D) and (Table 1), the cluster 1, including (c) 2-nitro-ethanol, (d) 1-methoxy-2-propanol, (e) 3-methyl-2-butanol, 2, (b) 4-dimethyl-heptane, (p) tetradecane and (q) 2-hexyl-1-decanol, represented the volatile compounds generated during the storage. The cluster 2, including (b) 4-penten-2-ol, (l) benzaldehyde, (n) nonanal, and (r) oleic acid, represented the volatile compounds the concentrations of which increased during the storage. The concentrations of these volatile compounds had significant changes during the storage (Figure 5C). The cluster 3 included the chemicals of (a) 2-methoxy-ethanol, (f) 2-pentano, (g) paraldehyde, (h) butanoic acid, (i) hexanal, (k) o-xylene, (m) 2-ethyl-1-hexanol, and (o) naphthalene, which disappeared during the storage.

As shown in (Figure 5C), the concentration of 18 volatile compounds which had significant changes during the moisture adsorption. It was observed that 4 volatile compounds increased after moisture absorption. The concentrations of (b) 4-penten-2-ol, (l) benzaldehyde, (n) nonanal, and (r) oleic acid increased to 26.22 ± 0.30, 7.10 ± 0.20, 4.70 ± 0.20, and 5.61 ± 0.20 from 22.21 ± 0.20, 2.75 ± 0.30, 1.47 ± 0.30, and 3.32 ± 0.30 ng/g, respectively. Meanwhile, the changes of (c) 2-nitro-ethanol, (d) 1-methoxy-2-propanol, (e) 3-methyl-2-butanol, and (f) 2-pentanol were sizable. The principal component analysis (PCA) results revealed that the differences of the volatile compounds are shown in (Figure 5E). The PC1 and PC2 accounted for 97.43% and 2.44% of the total peak areas’ variances, respectively. Among all the volatile compounds, the PC1 was accounted for its positive part by 10 volatile compounds. The concentration of the right volatile compounds increased or generated during the storage. About its negative part, the PC1 was explained by other compounds. The concentration of the left volatile compounds decreased during the storage. Meanwhile, the PC2 explained its positive part by 17 of total compounds and on its negative part by 1 volatile compound. These revealed that the samples generated some reactions in high environmental humidity. It was reported that the nonanal possessed a lipid flavor [48]. Meanwhile, the changes of alcohols were large after moisture absorption. Some alcohols derived from the oxidation of materials [17,48]. The action of oxidation or microorganisms might induce peptides, amino acids, and other compounds to degrade [10]. Meanwhile, the water combined with peptides might influence the release of volatile compounds. Therefore, the moisture absorption affected the volatile compounds of peptides. The moisture and temperature are also significant factors about flavor formation during the storage [49]. This study monitored the changes of volatile compounds at 75% RH and 25 °C. The peptides generated some volatile components and certain volatile components disappeared in this process. In brief, the moisture absorption affected the quality of SHs flour. The effects of the temperature and moisture on volatile components of SHs flour in the storage are still unclear. The mechanism of their formation will require further study.

CAS represents the registration number of chemical substances by Chemical Abstracts Service. GC-MS means the volatile has the MS fragments matching with the result of searching NIST MS library. RI^a^ was reported in the LRI database and the compound match is the more accurate when the R. match value is the higher. RI^b^ was obtained by PT-GC-MS using a HP-5MS Ultra Inert column with some alkanes between C8 and C20.

## 4. Conclusions

The moisture absorption affected the chemical properties of the SHs flour. DVS results showed that the moisture absorption rate of SHs flour could reach a maximum of 88.93% after moisture absorption. It could be seen from the LF-NMR results (Figure 2E) that the distribution of transverse relaxation times of SHs flour was changed during moisture absorption and the bound and immobilized water accounted for the majority of total water. Bound water and immobilized water transformed into each other. The moisture absorption reduced DPPH and OH radicals scavenging activity and the vibrations of the MIR absorbance peaks were changed. Meanwhile, the volatile compounds of SHs flour were changed during the process of the storage. The quality of SHs flour was affected by moisture absorption. Thus, the moisture absorption of hydrolysate flour should be prevented during the storage, such as nitrogen-filled packaging. Meanwhile, the quality assurance of functional hydrolysate needs to be further investigated.

## Figures and Tables

**Figure 1 foods-10-01137-f001:**
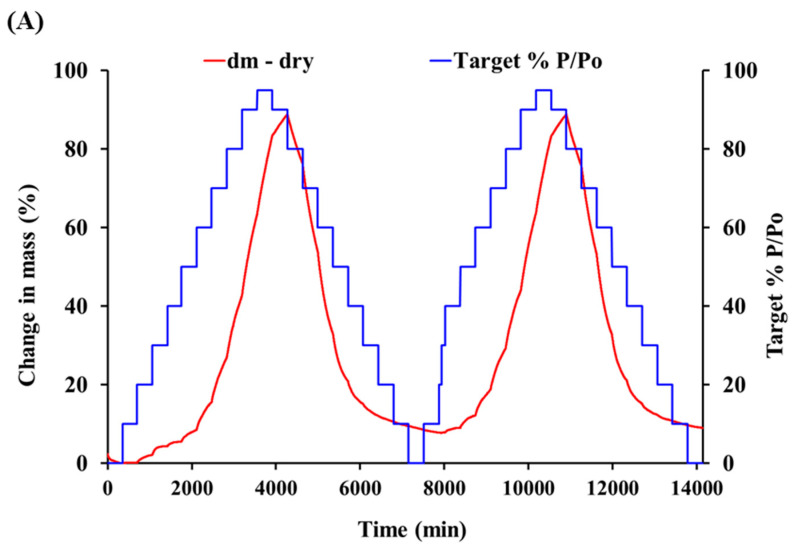
Dynamic vapor sorption measurements (from 0% to 95% RH) of SHs flour: (**A**) Water vapor sorption kinetic. (**B**) Changes of appearance shape. (**C**) Isotherm plot for SHs flour at 25 °C.

**Figure 2 foods-10-01137-f002:**
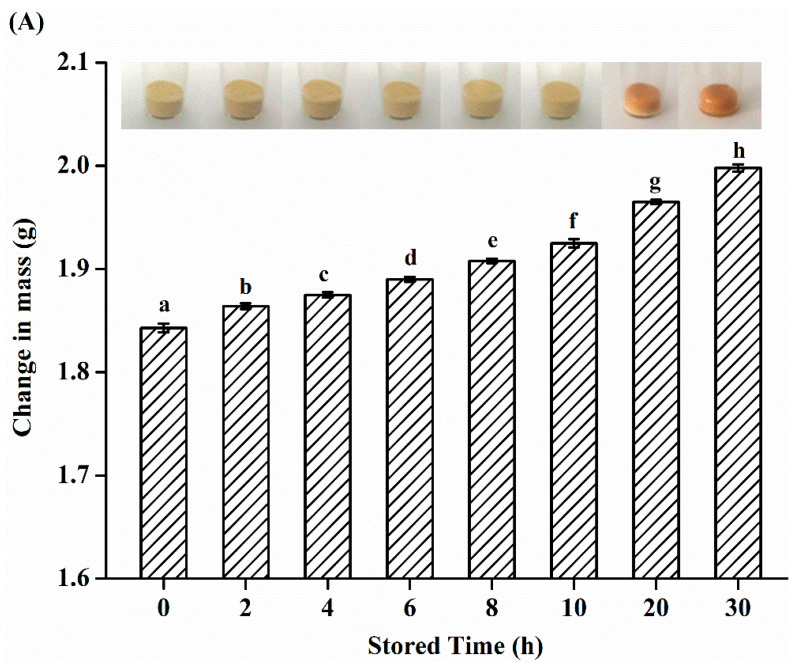
(**A**) The changes of appearance shape and weight of SHs flour absorbing the moisture at 25 °C and RH 75% at different storage times. (**B**) MRI pseudo-color images of SHs flour absorbing the moisture at 25 °C and RH 75% at different storage times. (**C**) LF-NMR *T*_2_ relaxation time distribution curves of SHs flour under the condition of 25 °C and RH 75% at different storage times. (**D**) Changes of water distribution in SHs flour during the process of storage. (**E**) Simplified models to illustrate the three types of water distribution in the SHs flour in the system of moisture absorption. The SHs-0 h, SHs-2 h, SHs-4 h, SHs-6 h, SHs-8 h, SHs-10 h, SHs-20 h, and SHs-30 h respectively represent the SHs flour stored for 0, 2, 4, 6, 8, 10, 20, and 30 h.

**Figure 3 foods-10-01137-f003:**
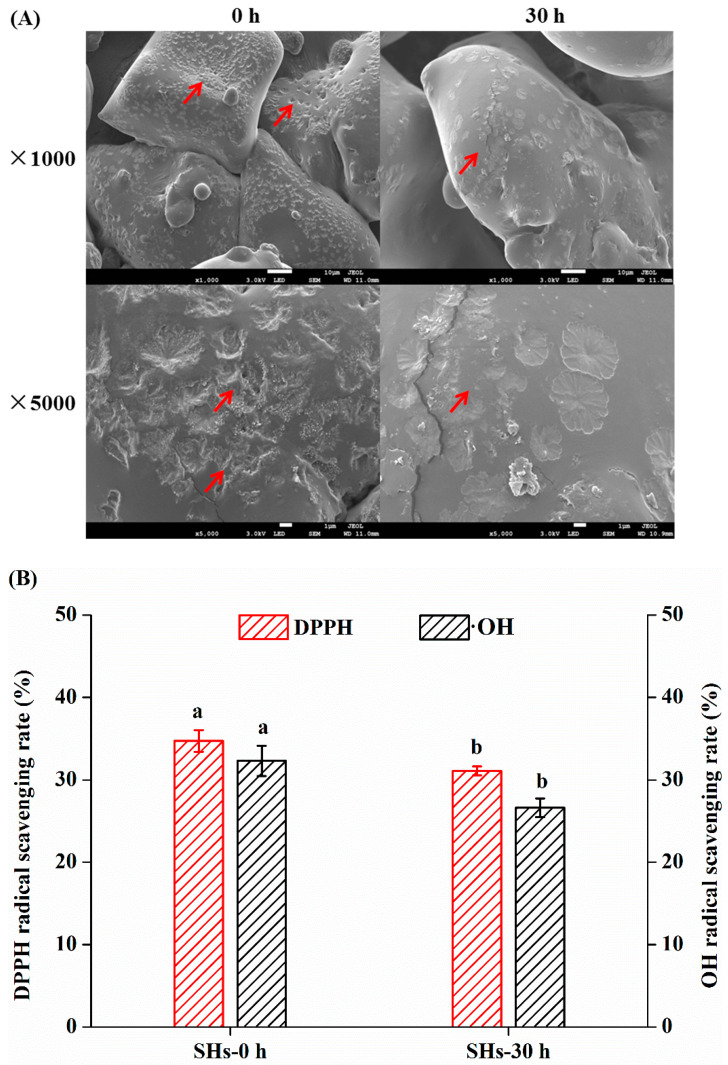
(**A**) Surface structure changes observed by SEM (1000× and 5000×). (**B**) The DPPH and OH radical scavenging rate of SHs-0 h and SHs-30 h. (**C**) The EPR spectra of DPPH radical scavenging ability. (**D**) The EPR spectra of OH radical scavenging ability.

**Figure 4 foods-10-01137-f004:**
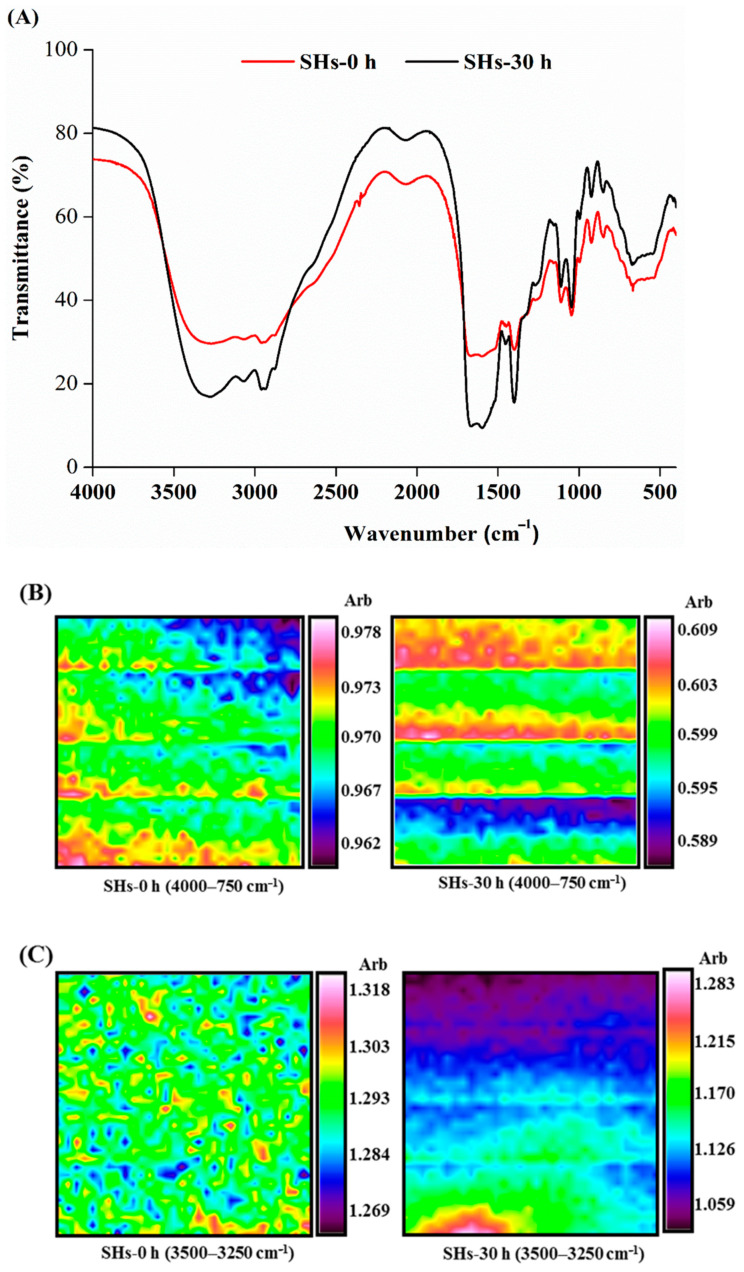
(**A**) Mid-infrared spectra of SHs-0 h and SHs-30 h. (**B**–**E**) Infrared microimaging spectra at 4000–750 cm^−1^, 3500–3250 cm^−1^, 1700–1600 cm^−1^ and 1560–1535 cm^−1^ of SHs-0 h and SHs-30 h.

**Figure 5 foods-10-01137-f005:**
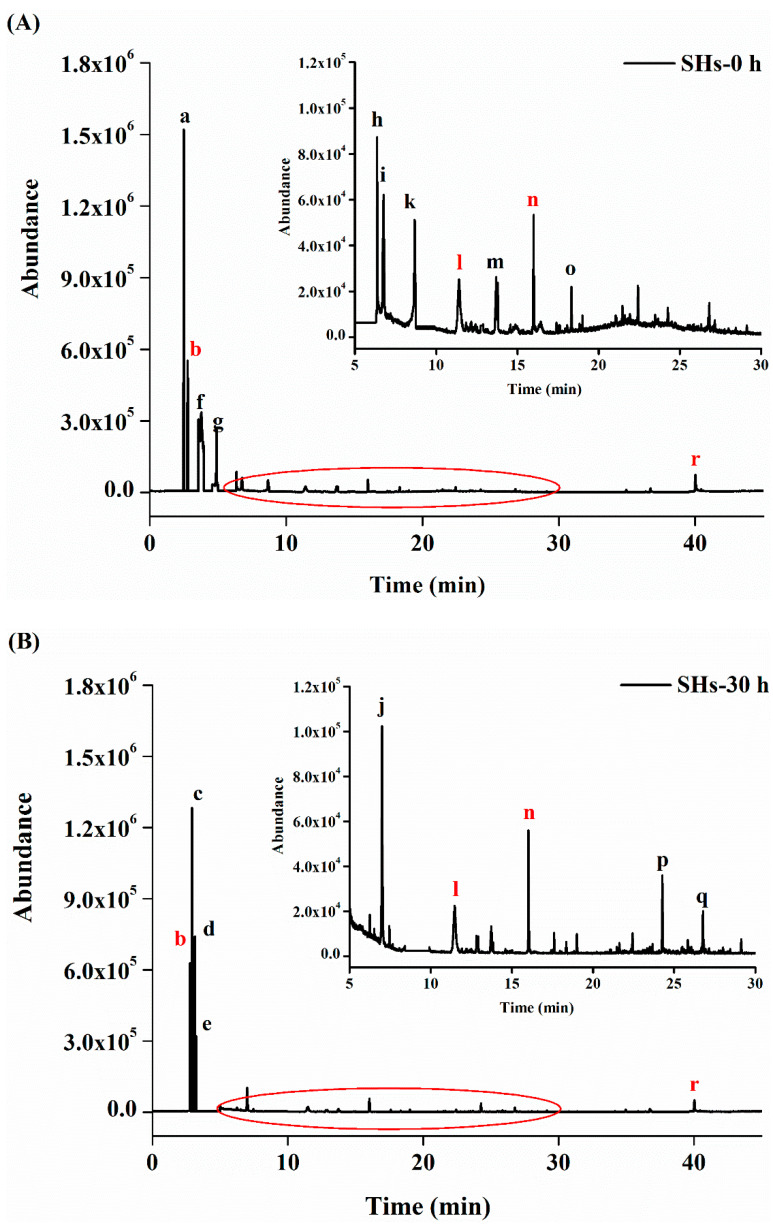
Chromatogram of SHs flour after PT-GC-MS analysis (Note: The letters refer to the volatile compounds listed in (Table 1). The red letters represented the volatile compounds always existed in SHs-0 h and SHs-30 h. The black letters represented the volatile compounds disappeared or generated during moisture absorption): (**A**) Chromatogram of SHs-0 h (**B**) Chromatogram of SHs-30 h. (**C**) The concentration of volatile compounds existed in SHs-0 h and SHs-30 h. (**D**) Hierarchical cluster analysis of SHs-0 h and SHs-30 h was analyzed basing on the standardized peak areas of volatile compounds. (**E**) The principal component analysis (PCA) scores plot.

**Table 1 foods-10-01137-t001:** Identification of the volatile compounds of SHS-0 h and SHS-30 h.

Peak No.	Time	Compounds Volatile	CAS	Molecule Fomula	RI^a^	RI^b^	Characteristic Fragment
a	2.494	2-Methoxy-ethanol	109-86-4	C_3_H_8_O_2_	621.9	624	14, 11, 29, 57
b	2.785	4-Penten-2-ol	625-31-0	C_5_H_10_O	643.4	647	45
c	2.922	2-Nitro-ethanol	625-48-9	C_2_H_5_NO_3_	653.9	658	43, 45
d	3.128	1-Methoxy-2-propanol	107-98-2	C_4_H_10_O_2_	669.5	661	43, 45, 47
e	3.226	3-Methyl-2-butanol	598-75-4	C_5_H_12_O	676.6	674	45, 55, 73
f	3.779	2-Pentanol	6032-29-7	C_5_H_12_O	709.2	703	43, 45
g	4.892	Paraldehyde	123-63-7	C_6_H_12_O_3_	753.8	755	43
h	6.365	Butanoic acid	107-92-6	C_4_H_8_O_2_	809.6	805	41, 60
i	6.763	Hexanal	66-25-1	C_6_H_12_O	821.6	800	44, 56, 72
j	6.999	2,4-Dimtthyl-heptane	2213-23-2	C_9_H_2_0	829.1	821	43, 57, 71
k	8.677	o-Xylene	95-47-6	C_8_H_10_	881.1	887	91, 106
l	11.409	Benzaldehyde	100-52-7	C_7_H_6_O	965.2	962	51, 77, 106
m	13.692	2-Ethyl-1-hexanol	104-76-7	C_8_H_18_O	1035.3	1030	41, 57, 70
n	16.000	Nonanal	124-19-6	C_9_H_18_O	1106.4	1104	41, 57, 70
o	18.329	Naphthalene	91-20-3	C_10_H_8_	1184.1	1182	102, 128
p	24.277	Tetradecane	629-59-4	C_14_H_3_O	1400.5	1400	43, 57, 71
q	26.769	2-hexyl-1-decanol	2425-77-6	C_16_H_34_O	1500.5	1504	43, 57, 71, 85
r	40.022	Oleic acid	112-80-1	C_18_H_34_O_2_	2141.1	2141	43, 55, 69, 83

## Data Availability

The data showed in this study are contained within the article.

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
