# Peer review of "Proton Dynamics of Water Diffusion in Shrimp Hydrolysates Flour and Effects of Moisture Absorption on Its Properties"

_foods, 2021, doi:10.3390/foods10051137_

Round 1

Reviewer 1 Report

This is an interesting paper and I would like the authors comments better on theri results.

I would like to ask to the authors to better develop the discussion, adding new and more updated references and then the paper can be considered after minor changes and among those a revision of the English language would help.

Author Response

Dear editor,

We acknowledge the thoughtful suggestions of the Referees, which helped in shaping of this paper: Proton dynamics of water diffusion in shrimp hydrolysates flour and effects of moisture absorption on its properties” (foods-1190080). All recommendations and specific comments were taken into account and the paper was modified accordingly. Besides, we also made some additions to the manuscript and self-corrected on some minor errors, which were not commented by reviewer.

This letter sent to the Editor-in-Chief indicated that we have closely revised the manuscript. And revised portion are underlined in red. If you have any questions, please contact us without hesitate. Thank you very much.

Yours sincerely,

*Corresponding author:

Professor Songyi Lin

No. 1 Qinggongyuan, Ganjingzi District

National Engineering Research Center of Seafood,

School of Food Science and Technology,

Dalian Polytechnic University, Dalian 116034, P.R. China

Tel.: +86 18840821971;

Fax: +86 411 86318655;

E-mail: linsongyi730@163.com

Reviewer #1:

We acknowledge for your advice and guidance on our manuscript. We authors here want to express our deep appreciation. We have revised our manuscript closely and carefully, the revised parts are listed below.

Comment: This is an interesting paper and I would like the authors comments better on theri results. I would like to ask to the authors to better develop the discussion, adding new and more updated references and then the paper can be considered after minor changes and among those a revision of the English language would help.

Answer: Thank you for your valuable comment. According to your comment, we have made the revision. We have revised the results discussion section. Meanwhile, we added many references in results discussion section. We have revised our research as follow:

Page 1 Line 31-32: Food proteins, especially hydrolyzed peptides, have important physiological activities including immunomodulatory, anticancer, and antioxidant [1-3].

Page 2 Line 59-61: Low-field nuclear magnetic resonance (LF-NMR) was used to monitor the water distribution and understand interactions between water and peptides or proteins by monitoring the proton dynamics [12-14].

Page 4 Line 183-188: The value is defined as the equilibrium moisture content. The DVS was used to evaluate the hygroscopicity of SHs flour. The water sorption kinetic curve was shown in (Figure 1A). The change of water content was presented by the change in the weight of the SHs flour. When a new target RH value was set, the water content of the SHs flour gradually changed during the process of adsorption and desorption until the state of the sorption reached a new state.

Page 5 Line 223-226: Whether the hysteresis behavior of SHs flour was evident or not depends on the amino acid residues of peptide flour, respectively [9]. A lot of amino acids might exist in the peptide sequence of SHs flour, which facilitated the hysteresis of desorption.

Page 8 Line 272-274: However, the detailed information about changes of the proton signal cannot be observed from the MRI [25]. The changes of the proton signal were further studied by relaxation time distribution curves of LF-NMR.

Page 8 Line 275-277: In general, the water dynamics in food were obtained by LF-NMR technology [25, 26]. Many researches had no in-depth understanding of the proton dynamics about interaction of the water and peptides.

Page 8 Line 285-288: Three hydrogen proton fractions in the process of moisture absorption were observed, revealing that three categories of water were observed in the SHs flour [30]. We found that the SHs flour had varying relaxation times and amplitudes during the process of storage.

Page 8 Line 281-284: Firstly, T21, component less than 10 ms,includes the protein-associated water, secondly, T22, the second component ranging from 10-100 ms, referring to the immobilized water and thirdly, T23, a component ranging from 100-200 ms referring to the free water [13, 27-29].

Page 8 Line 297-300: Study have shown that the binding force between material and bound water was strengthened when T22 became smaller [33]. As shown in (Figure 2C and Figure 2D), the content of bound and immobilized water gradually increased throughout the experiment. In the initial stage of moisture absorption, the immobilized water migrated to the bound water.

Page 8 Line 307-315: There were many intermolecular effects such as hydrogen bonds, electrostatic interactions, or electron density repulsions in protein structure [34]. Study has shown that the behaviors of moisture absorption were related to the polar groups of the peptides [9]. The interactions of water and peptides were related to these intermolecular effects. When proteins were enzymolyzed, amino acid residues were produced [35]. The amino acid residues might belong to the possible hydrogen bonding sites which induced the interactions between the peptides and moisture [10]. These sites probably provided water molecules with attachment sites and increased the chances of the interaction between water and peptides during the process of moisture absorption.

Page 11 Line 357-362: Compared with their crystalline solids, amorphous solids can absorb more water, which moisture absorption might result in particle agglomeration and flour caking [40]. The hygroscopicity caused morphological changes of peptide, which was changed from an amorphous structure into different agglomerates [41]. This was consistent with our findings that moisture absorption led to flour agglomeration and caking and lost the porous structure.

Page 11 Line 377-383: The reductive groups in hydrolysate flour were oxidized by oxygen and the antioxidant ability of SHs flour showed a certain degree of decline. The antioxidant activity of peptides was closely related to the exposure of amino acid residues [43]. During storage, amino acid residues may interact with water, which affected the exposure of amino acid residues and led to a decrease in antioxidant activity. These indicated that properties of SHs flour were not stable in a high humidity environment. However, the mechanism needs to be further studied.

Page 16 Line 460-476: The SHs-30 h was detected 6 new volatile compounds including (c) 2-nitro-ethanol, (d) 1-methoxy-2-propanol, (e) 3-methyl-2-butanol, 2, (b) 4-dimethyl-heptane, (p) tetradecane and (q) 2-hexyl-1-decanol. Cluster analysis of SHs flour’s volatile compounds was conducted according to the standardized peak areas of volatiles. The classification of each flavor component in the SHs-0 h and SHs-30 h flour can be clearly observed. As shown in (Figure 5D) and (Table 1), the cluster 1, including (c) 2-nitro-ethanol, (d) 1-methoxy-2-propanol, (e) 3-methyl-2-butanol, 2, (b) 4-dimethyl-heptane, (p) tetradecane and (q) 2-hexyl-1-decanol, represented the volatile compounds generated during the storage. The cluster 2, including (b) 4-penten-2-ol, (l) benzaldehyde, (n) nonanal, and (r) oleic acid, represented the volatile compounds the concentrations of which increased during the storage. The concentrations of these volatile compounds had significant changes during the storage. The cluster 3 included the chemicals of (a) 2-methoxy-ethanol, (f) 2-pentano, (g) paraldehyde, (h) butanoic acid, (i) hexanal, (k) o-xylene, (m) 2-ethyl-1-hexanol, and (o) naphthalene, which disappeared during the storage.

Page 16 Line 475-502: As shown in (Figure 5C), the concentration of 18 volatile compounds which had significant changes during the moisture adsorption. It was observed that 4 volatile compounds increased after moisture absorption. The concentrations of (b) 4-penten-2-ol, (l) benzaldehyde, (n) nonanal, and (r) oleic acid increased to 26.22 ± 0.30, 7.10 ± 0.20, 4.70 ± 0.20, and 5.61 ± 0.20 from 22.21 ± 0.20, 2.75 ± 0.30, 1.47 ± 0.30, and 3.32 ± 0.30 ng/g, respectively. Meanwhile, the changes of (c) 2-nitro-ethanol, (d) 1-methoxy-2-propanol, (e) 3-methyl-2-butanol, and (f) 2-pentanol were sizable. The principal component analysis (PCA) results revealed that the differences of the volatile compounds are shown in (Figure 5E). The PC1 and PC2 accounted for 97.43% and 2.44% of the total peak areas’ variances, respectively. Among all the volatile compounds, the PC1 was accounted for its positive part by 10 volatile compounds. The concentration of the right volatile compounds increased or generated during the storage. About its negative part, the PC1 was explained by other compounds. The concentration of the left volatile compounds decreased during the storage. Meanwhile, the PC2 explained its positive part by 17 of total compounds and on its negative part by 1 volatile compound. These revealed that the samples generated some reactions in high environmental humidity. It was reported that the nonanal possessed a lipid flavor [53]. Meanwhile, the changes of alcohols were large after moisture absorption. Some alcohols derived from the oxidation of materials [17, 53]. The the action of oxidation or microorganisms might induce peptides, amino acids, and other compounds to degrade [10]. Meanwhile, the water combined with peptides might influence the release of volatile compounds. Therefore, the moisture absorption affected the volatile compounds of peptides. The moisture and temperature are also significant factors about flavor formation during the storage [54]. This study monitored the changes of volatile compounds at 75% RH and 25 °C. The peptides generated some volatile components and certain volatile components disappeared in this process. In brief, the moisture absorption affected the quality of SHs flour. The effects of the temperature and moisture on volatile components of SHs flour in the storage are still unclear. The mechanism of their formation will require further study.

References:

Page 22 Line 547-548: [3] Thaha, A., Wang, B., Chang, Y., Hsia, S., Huang, T., Shiau, C. YHwang D. & Chen T. Food-derived bioactive peptides with antioxidative capacity, Xanthine Oxidase and Tyrosinase Inhibitory Activity, 2021, 9.

Page 22 Line 568-569: [14] Wang, S., Lin, R., Cheng, S., Wang, Z., & Tan, M. Assessment of water mobility in surf clam and soy protein system during gelation using lf-nmr technique. Foods, 2020, 9(2), 213.

Page 23 Line 593-595: [25] Huang L., Song Y., Kamal T., Li , Xia K., Lin Z., Qi L., Cheng S., Zhu B. and Tan M. A non-invasive method based on low-field NMR to analyze the quality changes in caviar from hybrid sturgeon (Huso dauricus, Acipenser schrenckiid). Journal of Food Processing and Preservation, 2017:e13256.

Page 23 Line 604-605: [30] Bi, J., Yong, L., Cheng, S., Dong, X., Kamal, T., & Zhou, D. Changes in body wall of sea cucumber (stichopus japonicus ) during a two-step heating process assessed by rheology, lf-nmr, and texture profile analysis. Food Biophysics, 2016, 11, 257-265.

Page 23 Line 611-612: [33] Cheng, S., Tang, Y., Tan, Z., Song, Y., & Tan, M. An approach for monitoring the dynamic states of water in shrimp during drying process with lf-nmr and mri. Drying Technology, 2018, 36.

Page 23 Line 628-629: [41] Wang, K., Sun, N., Li D. Cheng S., Liang R., and Lin S. Enzyme-controlled hygroscopicity and proton dynamics in sea cucumber (stichopus japonicus) ovum peptide powders. Food Research International, 2018, 112, 241-249.

Other modifications marked in the text:

Page 18 Line 505-509: CAS represents the registration number of chemical substances by Chemical Abstracts Service. GC-MS means the volatile has the MS fragments matching with the result of searching NIST MS library. RIa was reported in the LRI database and the compound match is the more accurate when the R. match value is the higher. RIb was obtained by PT-GC-MS using a HP-5MS Ultra Inert column with some alkanes between C8 and C20.

Reviewer 2 Report

Dear Author

The fig 5D and 5E seems irrelevant as it doesnt add any meaningful interpretation for discussion. 

Methods section needs elaboration appropriately in connection with results section.

Author Response

Dear editor,

We acknowledge the thoughtful suggestions of the referees, which helped in shaping of this paper: Proton dynamics of water diffusion in shrimp hydrolysates flour and effects of moisture absorption on its properties” (foods-1190080). All recommendations and specific comments were taken into account and the paper was modified accordingly. Besides, we also made some additions to the manuscript and self-corrected on some minor errors, which were not commented by reviewer.

This letter sent to the Editor-in-Chief indicated that we have closely revised the manuscript. And revised portion are underlined in red. If you have any questions, please contact us without hesitate. Thank you very much.

Yours sincerely,

*Corresponding author:

Professor Songyi Lin

No. 1 Qinggongyuan, Ganjingzi District

National Engineering Research Center of Seafood,

School of Food Science and Technology,

Dalian Polytechnic University, Dalian 116034, P.R. China

Tel.: +86 18840821971;

Fax: +86 411 86318655;

E-mail: linsongyi730@163.com

Reviewer #2:

Thanks so much for your advice and guidance on our manuscript. We authors here want to express our deep appreciation. We have revised our manuscript closely and carefully, the revised parts are listed below.

Comment 1: The Fig. 5D and 5E seems irrelevant as it doesnt add any meaningful interpretation for discussion.

Answer: Thanks for your suggestion. We authors have revised this part and adjusted the writing logic of the article. The corrections are as follow:

Page 16 Line 447-474: Volatile compounds in SHs flour were identified according to the matched retention index (RI). Most volatile compounds were found in databases, which could represent the authority of the identification. Cyclohexanone served as an internal standard and the peak areas were standardized. The characteristic chromatograms of the SHs-0 h and SHs-30 h are shown in (Figure 5A and Figure 5B), respectively. As shows in (Table 1), a total of 18 volatile compounds with Chemical Abstracts Service (CAS) registry numbers were identified in the SHs-0 h and SHs-30 h. The volatile compounds were classified according to their structure, including 7 alcohols, 4 aldehydes, 4 arenes, 2 acids, and 1 ester. The SHs-0 h and SHs-30 h contained 12 and 10 major volatile components, respectively. There were four kinds of volatile compounds in SHs-0 h and SHs-30 h including (b) 4-penten-2-ol, (l) benzaldehyde, (n) nonanal, and (r) oleic acid. After SHs flour was stored for 30 h, eight volatile compounds disappeared, which included (a) 2-methoxy-ethanol, (f) 2-pentanol, (g) paraldehyde, (h) butanoic acid, (i) hexanal, (k) o-xylene, (m) 2-ethyl-1-hexanol, and (o) naphthalene. The SHs-30 h was detected 6 new volatile compounds including (c) 2-nitro-ethanol, (d) 1-methoxy-2-propanol, (e) 3-methyl-2-butanol, 2, (b) 4-dimethyl-heptane, (p) tetradecane and (q) 2-hexyl-1-decanol. Cluster analysis of SHs flour’s volatile compounds was conducted according to the standardized peak areas of volatiles. The classification of each flavor component in the SHs-0 h and SHs-30 h flour can be clearly observed. As shown in (Figure 5D) and (Table 1), the cluster 1, including (c) 2-nitro-ethanol, (d) 1-methoxy-2-propanol, (e) 3-methyl-2-butanol, 2, (b) 4-dimethyl-heptane, (p) tetradecane and (q) 2-hexyl-1-decanol, represented the volatile compounds generated during the storage. The cluster 2, including (b) 4-penten-2-ol, (l) benzaldehyde, (n) nonanal, and (r) oleic acid, represented the volatile compounds the concentrations of which increased during the storage. The concentrations of these volatile compounds had significant changes during the storage. The cluster 3 included the chemicals of (a) 2-methoxy-ethanol, (f) 2-pentano, (g) paraldehyde, (h) butanoic acid, (i) hexanal, (k) o-xylene, (m) 2-ethyl-1-hexanol, and (o) naphthalene, which disappeared during the storage.

Page 17 Line 475-502: As shown in (Figure 5C), the concentration of 18 volatile compounds which had significant changes during the moisture adsorption. It was observed that 4 volatile compounds increased after moisture absorption. The concentrations of (b) 4-penten-2-ol, (l) benzaldehyde, (n) nonanal, and (r) oleic acid increased to 26.22 ± 0.30, 7.10 ± 0.20, 4.70 ± 0.20, and 5.61 ± 0.20 from 22.21 ± 0.20, 2.75 ± 0.30, 1.47 ± 0.30, and 3.32 ± 0.30 ng/g, respectively. Meanwhile, the changes of (c) 2-nitro-ethanol, (d) 1-methoxy-2-propanol, (e) 3-methyl-2-butanol, and (f) 2-pentanol were sizable. The principal component analysis (PCA) results revealed that the differences of the volatile compounds are shown in (Figure 5E). The PC1 and PC2 accounted for 97.43% and 2.44% of the total peak areas’ variances, respectively. Among all the volatile compounds, the PC1 was accounted for its positive part by 10 volatile compounds. The concentration of the right volatile compounds increased or generated during the storage. About its negative part, the PC1 was explained by other compounds. The concentration of the left volatile compounds decreased during the storage. Meanwhile, the PC2 explained its positive part by 17 of total compounds and on its negative part by 1 volatile compound. These revealed that the samples generated some reactions in high environmental humidity. It was reported that the nonanal possessed a lipid flavor [53]. Meanwhile, the changes of alcohols were large after moisture absorption. Some alcohols derived from the oxidation of materials [17, 53]. The the action of oxidation or microorganisms might induce peptides, amino acids, and other compounds to degrade [10]. Meanwhile, the water combined with peptides might influence the release of volatile compounds. Therefore, the moisture absorption affected the volatile compounds of peptides. The moisture and temperature are also significant factors about flavor formation during the storage [54]. This study monitored the changes of volatile compounds at 75% RH and 25 °C. The peptides generated some volatile components and certain volatile components disappeared in this process. In brief, the moisture absorption affected the quality of SHs flour. The effects of the temperature and moisture on volatile components of SHs flour in the storage are still unclear. The mechanism of their formation will require further study.

Comment 2: Methods section needs elaboration appropriately in connection with results section.

Answer: Thanks for your suggestion. We have made the revisions about the method sections. The corrections are as follow:

Page 3 Line 97-99: The LF-NMR experiment was performed according to the method of Yang et al. with some modifications [8]. The water dynamics and distribution of SHs flour were monitored by LF-NMR equipment. The NMI20-030H-INMR analyzer (Niumag Electric Corp., Shanghai, China) was used in the LF-NMR experiment.

Page 3 Line 124-126: The effects of moisture absorption on morphological changes of SHs flour were evaluated by SEM. SEM experiment was performed according to the previous method described by Zhao et al. [15].

Page 3 Line 137-139: The effects of moisture absorption on secondary structure were evaluated by MIR spectroscopy analysis. The MIR analysis was performed according to the method of Lin et al. [18].

Page 4 Line 143-147: Infrared microscopic imaging technology was used to further study the effects of moisture absorption on secondary structure and can be used to measure the functional group information and spatial distribution information of the molecular structure of proteins and other samples. Infrared microscopic imaging was performed according to the method of Bonwell et al. with some modifications [19].

Page 4 Line152-153: Volatiles components in SHs flour were analyzed through purge and trap coupled with gas chromatography-mass spectrometry (PT-GC-MS) (Agilent/Atomx/7890B/5977A), according to the method of PozoBayón et al. [20].

Other modifications marked in the text:

Page 18 Line 505-509: CAS represents the registration number of chemical substances by Chemical Abstracts Service. GC-MS means the volatile has the MS fragments matching with the result of searching NIST MS library. RIa was reported in the LRI database and the compound match is the more accurate when the R. match value is the higher. RIb was obtained by PT-GC-MS using a HP-5MS Ultra Inert column with some alkanes between C8 and C20.

Reviewer 3 Report

  This paper described that the effect of moisture on shrimp hydrolysates flour during storage is being investigated from multiple perspectives (DVS, LF-NMR, EPR, MIR, and GCMS analysis). This paper has been very well-organized and English is clear. The results are giving an interesting and valuable information. In addition, the authors' conclusions are very significant because of the multifaceted evaluation. However, there are some problems and flaws in presentation n.  I hope that my comments are very useful for the improvement of this research.

Comments

  • Abstract: Authors need to explain how you prepared SH-0 h and SHs-30 h. In addition, instrumental analysis needs to show what it is analyzing to reveal. For example, looking at the abstract alone, it is not clear what was used to analyze the volatile components. It would be better to know which instrument was used to analyze the item that was measured, even if you only look at this abstract.
  • L67: Since amino acids are mentioned in the discussion, it would be better to measure the amino acid composition of the flour. If you have information in the past literature, please indicate that.
  • L69: Please indicate the species of shrimp. Please also indicate which part of the shrimp is used.
  • L84-92: It is in bold, so please change it to a normal font.
  • L140-157: Shrimp hydrolysates flour is dried by freeze-drying. Therefore, I believe that many volatile components have been already volatilized. Therefore, I wonder if what authors are measuring here are the components generated during the GCMS analysis. I think this point needs to be considered.
  • L158-162: The method of significance test is not indicated. Please show this method.
  • L515: It is better to indicate ï½¥OH with OH.

Author Response

Dear editor,

We acknowledge the thoughtful suggestions of the Referees, which helped in shaping of this paper: Proton dynamics of water diffusion in shrimp hydrolysates flour and effects of moisture absorption on its properties” (foods-1190080). All recommendations and specific comments were taken into account and the paper was modified accordingly. Besides, we also made some additions to the manuscript and self-corrected on some minor errors, which were not commented by reviewer.

This letter sent to the Editor-in-Chief indicated that we have closely revised the manuscript. And revised portion are underlined in red. If you have any questions, please contact us without hesitate. Thank you very much.

Yours sincerely,

*Corresponding author:

Professor Songyi Lin

No. 1 Qinggongyuan, Ganjingzi District

National Engineering Research Center of Seafood,

School of Food Science and Technology,

Dalian Polytechnic University, Dalian 116034, P.R. China

Tel.: +86 18840821971;

Fax: +86 411 86318655;

E-mail: linsongyi730@163.com

Reviewer #3:

We acknowledge for your advice and guidance on our manuscript. We authors here want to express our deep appreciation. We have revised our manuscript closely and carefully, the revised parts are listed below.

This paper described that the effect of moisture on shrimp hydrolysates flour during storage is being investigated from multiple perspectives (DVS, LF-NMR, EPR, MIR, and GCMS analysis). This paper has been very well-organized and English is clear. The results are giving interesting and valuable information. In addition, the authors' conclusions are very significant because of the multifaceted evaluation. However, there are some problems and flaws in presentation n. I hope that my comments are very useful for the improvement of this research.

Comment 1: Abstract: Authors need to explain how you prepared SH-0 h and SHs-30 h. In addition, instrumental analysis needs to show what it is analyzing to reveal. For example, looking at the abstract alone, it is not clear what was used to analyze the volatile components. It would be better to know which instrument was used to analyze the item that was measured, even if you only look at this abstract.

Answer: Thank you for your valuable comment. According to your comment, we have made the revision. We have revised our research as follow:

Page 1 Line 11-18: Moisture absorbed into shrimp hydrolysates (SHs) flour profoundly affected its properties. The unstored hydrolysate flour was named SHs-0h and SHs stored for 30 h at 25 °C and 75% relative humidity was named SHs-30. During the process of storage, the moisture dynamics in SHs flour were investigated by dynamic vapor sorption (DVS) and low-field nuclear magnetic resonance (LF-NMR). The effects of moisture absorption on the radicals scavenging rates of SHs flour were evaluated by electron paramagnetic resonance (EPR), The effects of moisture absorption on secondary structure were studied by mid-infrared (MIR) spectroscopy and infrared microimaging spectroscopy. The changes of volatile components were monitored by purge and trap coupled with gas chromatography-mass spectrometry (PT-GC-MS).

Comment 2: Since amino acids are mentioned in the discussion, it would be better to measure the amino acid composition of the flour. If you have information in the past literature, please indicate that.

Answer: Thanks for your suggestion. We apologize for the inaccuracy of our description about amino acids and have deleted inappropriate descriptions. Relevant study has shown that amino acid residues were produced when proteins were enzymolyzed. Thus, the discussion about amino acid residues is preserved. The corrections are as follow:

Page 8 Line 308-313: Study has shown that the behaviors of moisture absorption were related to the polar groups of the peptides [9]. The interactions of water and peptides were related to these intermolecular effects. When proteins were enzymolyzed, amino acid residues were produced [35]. The amino acid residues might belong to the possible hydrogen bonding sites which induced the interactions between the peptides and moisture [10].

Page 11 Line 378-381: The antioxidant activity of peptides was closely related to the exposure of amino acid residues [43]. During storage, amino acid residues may interact with water, which affected the exposure of amino acid residues and led to a decrease in antioxidant activity.

Comment 3: Please indicate the species of shrimp. Please also indicate which part of the shrimp is used.

Answer: Thank you for your valuable comment. We added a description of the part of the shrimp used in the experiment. The shrimp meat of freshwater shrimp was used in experimental research. We authors have revised this part. The corrections are as follow:

Page 2 Line 72-73: Two kilograms of shrimp meat of freshwater shrimp was purchased at a local fish market in Dalian (China).

Comment 4: It is in bold, so please change it to a normal font.

Answer: Thanks for your suggestion. We authors have revised this part. The corrections are as follow:

Page 2 Line 87-95: Moisture sorption experiment was performed according to the method at 25 °C by exposing the SHs flour to different values of RH within the range from 0 to 95% using the DVS (Surface Measurement Systems Ltd, London, UK) [16]. Approximately, 13 mg of SHs flour was used for each experiment. First, the SHs flour was dried by exposing it to dry nitrogen until a constant weight was reached. The RH was raised in RH steps of 10% from 0% to 90% RH and in RH steps of 5% from 90% to 95%. Then in reverse order to 0% RH and the experiment was conducted in two cycles. It was increased to the next target value if the sample mass rate with time was lower than 0.002% min-1 or the running time was over 360 min.

Comment 5: Shrimp hydrolysates flour is dried by freeze-drying. Therefore, I believe that many volatile components have been already volatilized. Therefore, I wonder if what authors are measuring here are the components generated during the GCMS analysis. I think this point needs to be considered.

Answer: Thanks for your kind suggestion and correction to our manuscript. We have clarified our research as follow:

Shrimp hydrolysates flour was dried by freeze-drying and many volatile components may have been already volatilized. We cannot avoid the loss of volatile components. Our purpose was to detect the difference between the volatiles components of shrimp hydrolysates flour stored for 30 h (SHs-30 h) and that of unstored shrimp hydrolysates flour (SHs-0 h). SHs-0 h and SHs-30 h flour were freeze-dried under the same conditions and prepared for the GCMS analysis in same conditions. Reproducibility tests were carried out and the volatile components of SHs-0h and SH-30h were different. Thus, the differences of volatile components of the SHs-0 h and SHs-30 h flour were more likely to generate during storage.

Comment 6: The method of significance test is not indicated. Please show this method.

Answer: Thanks for your kind suggestion to our manuscript. We supplemented the method of significance analysis. The corrections are as follow:

Page 4 Line 171-175: Statistical analyses were performed using the SPSS 21.0 version software (SPSS Inc., Chicago, IL, USA). The experimental results were shown as means ± standard deviation (SD). A Student’s t-test or one-way analysis of variance was used to analyze significant difference. Three replicated measured results were performed and a level of significance P < 0.05 was considered significant.

Comment 7: It is better to indicate ï½¥OH with OH.

Answer: Thanks for your suggestion. We authors have revised this part. We replaced ï½¥OH with OH. The corrections are as follow:

Page 1 Line 22-24: DPPH (31.09±0.54%) and OH (26.62±1.14%) radicals scavenging rates of SHs-30 h significantly reduced (P < 0.05) compared to that of SHs-0 h flour.

Page 3 Line 132-135: The antioxidant activity was measured by EPR spectrometer (Bruker, Karisruhe, Germany) and the DPPH and OH scavenging capacity were measured based on the method of Ding et al. [17]. The concentration of ultrafiltration and purified fractions were changed (1mg/mL).

Page 11 Line 368-371: The 1, 1-diphenyl-2-picrylhydrazyl (DPPH) and hydroxyl (OH) radical scavenging activity of SHs flour was measured and the EPR signal intensity was shown (Figure 3). SHs flour, the DPPH and OH radical scavenging activities were obtained through the integral of the peak.

Page 12 Line 384-386:                                                         

Page 13 Line 389-391: Figure 3. (A) Surface structure changes observed by SEM (1000× and 5000×). (B) The DPPH and OH radical scavenging rate of SHs-0 h and SHs-30 h. (C) The EPR spectra of DPPH radical scavenging ability. (D) The EPR spectra of OH radical scavenging ability.

Page 14 Line 433-435: Wang et al. reported that the OH radical scavenging capacity of sea cucumber peptide flour decreased after it was stored for 24 h [26], which was consistent with our result.

Page 22 Line 529-531: The moisture absorption reduced DPPH and OH radicals scavenging activity and the vibrations of the MIR absorbance peaks were changed.

Other modifications marked in the text:

Page 18 Line 505-509: CAS represents the registration number of chemical substances by Chemical Abstracts Service. GC-MS means the volatile has the MS fragments matching with the result of searching NIST MS library. RIa was reported in the LRI database and the compound match is the more accurate when the R. match value is the higher. RIb was obtained by PT-GC-MS using a HP-5MS Ultra Inert column with some alkanes between C8 and C20.
